# GHSR Deletion in β-Cells of Male Mice: Ineffective in Obesity, but Effective in Protecting against Streptozotocin-Induced β-Cell Injury in Aging

**DOI:** 10.3390/nu16101464

**Published:** 2024-05-13

**Authors:** Hye Won Han, Geetali Pradhan, Daniel Villarreal, Da Mi Kim, Abhishek Jain, Akhilesh Gaharwar, Yanan Tian, Shaodong Guo, Yuxiang Sun

**Affiliations:** 1Department of Nutrition, Texas A&M University, College Station, TX 77843, USA; hyeewoon@tamu.edu (H.W.H.);; 2USDA/ARS Children’s Nutrition Research Center, Department of Pediatrics, Baylor College of Medicine, Houston, TX 77030, USA; 3Interdepartmental Program in Translational Biology and Molecular Medicine, Baylor College of Medicine, Houston, TX 77030, USA; 4Department of Biomedical Engineering, College of Engineering, Texas A&M University, College Station, TX 77843, USA; 5Department of Physiology and Pharmacology, College of Veterinary Medicine and Biomedical Sciences, Texas A&M University, College Station, TX 77843, USA

**Keywords:** ghrelin, growth hormone secretagogue receptor (GHSR), aging, streptozotocin (STZ), pancreatic islets, β-cell

## Abstract

Insulin secretion from pancreatic β cells is a key pillar of glucose homeostasis, which is impaired under obesity and aging. Growth hormone secretagogue receptor (GHSR) is the receptor of nutrient-sensing hormone ghrelin. Previously, we showed that β-cell GHSR regulated glucose-stimulated insulin secretion (GSIS) in young mice. In the current study, we further investigated the effects of GHSR on insulin secretion in male mice under diet-induced obesity (DIO) and streptozotocin (STZ)-induced β-cell injury in aging. β-cell-specific-*Ghsr*-deficient (*Ghsr*-βKO) mice exhibited no glycemic phenotype under DIO but showed significantly improved ex vivo GSIS in aging. We also detected reduced insulin sensitivity and impaired insulin secretion during aging both in vivo and ex vivo. Accordingly, there were age-related alterations in expression of glucose transporter, insulin signaling pathway, and inflammatory genes. To further determine whether GHSR deficiency affected β-cell susceptibility to acute injury, young, middle-aged, and old *Ghsr*-βKO mice were subjected to STZ. We found that middle-aged and old *Ghsr*-βKO mice were protected from STZ-induced hyperglycemia and impaired insulin secretion, correlated with increased expression of insulin signaling regulators but decreased pro-inflammatory cytokines in pancreatic islets. Collectively, our findings indicate that β-cell GHSR has a major impact on insulin secretion in aging but not obesity, and GHSR deficiency protects against STZ-induced β-cell injury in aging.

## 1. Introduction

Obesity and aging are major risk factors for type 2 diabetes (T2D) and glycemic dysfunction [1,2]. Aging is often accompanied by obesity; while obesity and aging share common characteristics of inflammation and insulin resistance, each has their unique characteristics in glycemic regulation [2]. In the early stages of obesity, β cells compensate for insulin resistance by expanding β-cell mass and increasing insulin secretion; this process can induce β-cell stress, ultimately resulting in β-cell failure and the onset of T2D [2,3,4]. Obesity is accompanied by chronic inflammation characterized by elevated levels of pro-inflammatory cytokines such as tumor necrosis factor-alpha (TNF-𝛼), C-reactive protein (CRP), and interleukins (IL-1, IL-6, and IL-18), with strong correlations between these inflammatory markers and body fat [5,6]. Although β cells necessitate pro-inflammatory signaling to induce proliferation, an elevation of these signals can also accelerate β-cell dysfunction [3,7]. In fact, islets highly express the IL-1 receptor type 1 (IL-1R1), making them more sensitive to IL-1β and susceptible to obesity-induced inflammation [4,8]. The inhibition of IL-1β expression in β cells has been shown to have a protective effect against β-cell dysfunction [9]. 

Globally, one in five people older than 65 live with diabetes, and diabetes is projected to increase as the aging population increases [10]. This leads to a devastatingly diminished quality of life and a significant global healthcare burden. Evidence suggests that several factors such as systemic chronic inflammation, oxidative stress, and cellular senescence are associated with the pathophysiology of diabetes in the elderly [11], but the precise mechanisms underlying age-related T2D are not well understood. The function of pancreatic β cells declines with age, which leads to glucose intolerance and diabetes [12,13]. Age-associated impaired insulin secretion is characterized by reduced β-cell sensitivity and impaired insulin response to hyperglycemia [14,15]. Furthermore, the increased demand for insulin secretion due to insulin resistance in the elderly places an additional burden on the β cells, further exacerbating their dysfunction [16,17]. Due to the insulin-resistant state in the aging, their insulin levels are comparable to or slightly higher than their young controls [14]. This poses additional challenges for assessment of the insulin secretory function in aging subjects. When subjected to a hyperglycemic challenge, insulin levels are lower in the islets of aged mice and humans, indicating the negative impact of aging on insulin secretion [14,16]. These findings underscore the critical importance of preserving insulin secretory function in aging. Currently, the precise mechanisms underlying β-cell dysfunction in aging are not fully understood. 

The growth hormone secretagogue receptor (GHSR) is a seven-transmembrane G-protein-coupled receptor (GPCR) that serves as the only recognized receptor for the circulating orexigenic hormone ghrelin [18,19]. A wide variety of physiological functions of GHSR have been reported, including regulation of feeding behavior, modulation of energy metabolism, regulation of immune response, etc. [20,21]. GHSR is primarily expressed in the brain, but low levels are also found in peripheral tissues such as spleen, adrenal gland, stomach, gut, and pancreatic islets [22,23,24]. Studies of GHSR expression in pancreatic β cells have yielded inconsistent results, some suggest that GHSR is expressed in β cells, while other studies show GHSR is not expressed in β cells [25,26,27,28]. Nonetheless, we confirmed that GHSR is expressed in β cells through tracking the surrogate GFP reporter expression and successfully generated β-cell-specific GHSR-deleted mice (*MIP-Cre/ERT;Ghsr^f/f^* or *Ghsr*-βKO) [29]. Using this mouse model, we demonstrated that GHSR has a critical role in glucose homeostasis, regulating insulin secretion and systemic insulin sensitivity in young adult mice [29]. Interestingly, both β-cell-specific GHSR-deleted mice and the global knockout of GHSR showed improvements in insulin sensitivity [29,30]. Furthermore, GHSR expression has been shown to be upregulated in the β cells of T2D patients [31]. These results collectively suggest that GHSR in β cells may play an important role in glucose homeostasis and the pathogenesis of diabetes. 

Variable results have been reported for global GHSR deletion mice in DIO; some studies showed no difference in weight gain under a high-fat diet (HFD) [32,33], whereas others showed a decrease in weight gain [34]. GHSR is also known for its association with aging, and we previously reported that the ablation of GHSR suppresses age-related changes in adipose/hepatic inflammation and insulin resistance, and we recently reported that macrophage GHSR controls systemic inflammation and insulin resistance in DIO [35,36,37,38,39,40]. 

However, the role of β-cell GHSR under HFD-induced obesity and normal aging processes have not been thoroughly examined. To better understand the role of GHSR in β-cell function under obesity and aging, we examined the insulin secretion of β-cell-specific *Ghsr*-deficient (*Ghsr*-βKO) mice under HFD-induced obesity and during aging. Additionally, to investigate the impact of GHSR deficiency on β-cell resilience to acute inflammatory injury, we challenged young, middle-aged, or old *Ghsr*-βKO and their control mice with β-cell injury agent streptozotocin (STZ). The outcome measures were the following: body weight, fed blood glucose, plasma insulin, serum metabolic hormones and pro-inflammatory cytokines, glucose tolerance testing (GTT), in vivo and ex vivo GSIS, insulin tolerance testing (ITT), and gene expressions in islets. The primary outcome measures were GSIS and GTT.

## 2. Materials and Methods

### 2.1. Animals

All procedures were approved by the Institutional Animal Care and Use Committee (IACUC) at Baylor College of Medicine (Protocol AN-2770) or Texas A&M University (Protocol 2016-0292). Animal rooms had controlled temperature (23 ± 1 °C) and lighting cycle (12 h light–dark cycle). Where not specified, the mice were given free access to a regular diet (RD) and water. The standard laboratory RD was obtained from Harlan Teklad (2920X, Indianapolis, IN, USA) with the following composition: 60% from carbohydrates, 24% from protein, and 16% of calories from fat. This study used male mice only, with the age of the mice specified in the figure legends. Our *Ghsr^f/f^* mice were backcrossed to congenic C57/BL6J mice as we previously described [29]. *Ghsr^f/f^* mice were further crossbred with mice expressing Cre recombinase under tamoxifen-inducible mouse insulin promoter (*MIP-Cre/ERT*). Littermate *MIP-Cre/ERT;Ghsr^f/f^* (*Ghsr*-βKO) and *Ghsr^f/f^* control (CTRL) mice were used in this study. *Ghsr*-βKO and CTRL mice at 2–3 months of age were gavaged with tamoxifen to *Ghsr* deletion, and a 4 mg/200 μL dose of tamoxifen (T5648, Sigma-Aldrich, St. Louis, MO, USA) dissolved in peanut oil (P2144, Sigma-Aldrich) was given to each mouse every other day, 5 times. 

The exclusion criteria for the study included mice exhibiting evidence of unhealthy conditions such as visible wounds or significant weight loss. No mice in the study met the exclusion criteria. Throughout the experiment, general health conditions and weight changes were monitored. A rapid bodyweight reduction of 20% or more within a few days after the tamoxifen gavage, STZ injection, or metabolic tests was considered a humane endpoint; however, such occurrences were not observed in this study.

Whole-body composition analysis was conducted using the Echo MRI-100 whole-body composition analyzer (Echo Medical Systems, Houston, TX, USA), following the manufacturer’s instructions. For plasma collection prior to sacrifice, whole blood was obtained from the tail vein using EDTA capillary tubes (Drummond Scientific, Broomall, PA, USA). At termination, mice were exposed to 2 mL of isoflurane to anesthetize, then retro-orbital bleeding was conducted to collect ~0.5 mL of blood for serum. Cervical dislocation was performed afterward, and the pancreatic islets were collected for total RNA extraction or ex vivo GSIS. 

### 2.2. High-Fat-Diet (HFD) Feeding and Streptozotocin-Treated Regular Diet (RD)-Fed Mice 

HFD cohort: *Ghsr*-βKO and CTRL mice were fed with HFD containing 42% of calories from fat and 42.7% from carbohydrate (TD.88137, Envigo, Madison, WI, USA). This feeding regimen commenced at 8 weeks of age for 16 weeks, and both groups of mice had ad libitum access to food and water throughout the study period. Body weights, body composition, and blood glucose levels were measured at the start of the feeding study and then every 2 weeks until the end of the study. 

RD cohort: RD-fed mice at various ages were intraperitoneally (IP) injected with STZ (S0130-1G, Sigma Chemical Co., St. Louis, MO, USA) specifically to induce pancreatic β-cell injury. A single high dose of STZ (150 mg/kg body weight) was administered via IP injection. STZ was dissolved in Na citrate buffer (0.1 M, pH 4.2), which was prepared on the day of injection. Prior to STZ injection, the mice were fasted for 4 h. After the injection, the mice were supplemented with 10% sucrose overnight to prevent severe hypoglycemia. Blood glucose levels were measured using a glucometer (OneTouch Ultra, LifeScan, Milpitas, CA, USA). Every alternate day, 25 μL of blood was taken using EDTA capillary tubes, and plasma was collected for insulin measurement. On day 11 of the study, pancreatic islets and serum samples were collected for further analysis. The pancreatic islets were isolated following our previously published islet isolation protocol [41]. At termination, blood was collected via retroorbital bleeding for serum collection.

### 2.3. Glucose Tolerance Test and In Vivo Glucose-Stimulated Insulin Secretion

To conduct IPGTT, mice were fasted overnight for 16 h (from 5 PM to 9 AM) prior to the test. During the test, mice were transferred to individual cages with ad libitum access to water. A glucose solution (10% glucose in saline) was prepared on the day of the experiment and filtered (0.22 µm). Mice were administered the solution at a dose of 1 g of glucose per kg of body weight. Blood glucose was measured at the indicated times using a glucometer at 0, 15, 30, 60, and 120 min, and 25 µL of blood was collected at 0, 15, 30, and 60 min for insulin measurement. Plasma was obtained from the tail vein using EDTA capillary tubes (Drummond Scientific, Broomall, PA, USA). Plasma samples were stored at −80 °C until further analysis.

For the in vivo GSIS experiment, mice were fasted for 16 h (from 6 PM to 10 AM) prior to the test. After fasting, mice were IP injected with D-glucose (3 g per kg of body weight). Blood glucose was measured with a glucometer, and 25 µL samples of blood were collected from the tails at 0, 2, 5, 15, and 30 min after glucose injection. Plasma samples were obtained from the tail vein using EDTA capillary tubes for insulin measurement.

### 2.4. Ex Vivo Glucose-Stimulated Insulin Secretion

Mouse pancreatic islets were isolated following our previously published islet isolation protocol [41]. Briefly, a collagenase P (Sigma-Aldrich, St. Louis, MO, USA) solution was injected into the ampulla of Vater in the pancreas. The digested pancreas was collected and further digested in a 37 °C shaking water bath at a speed of 100 rpm for 12–14 min. After digestion, 10% fetal bovine serum (FBS) in ice-cold 1X HBSS was added to stop the digestion. This was followed by multi-step centrifugation, which included gradient separation using Histopaque-1077 (Sigma-Aldrich, St. Louis, MO, USA). The islets were subsequently hand-picked under a microscope and then incubated overnight at 37 °C in RPMI 1640 medium with 5.5 mM glucose. For the ex vivo GSIS assay, islets were either incubated with RPMI 1640 with 3.3 mM glucose for basal condition, or 22.2 mM glucose for hyperglycemic condition [30]. After stimulation, conditioned media were collected for insulin secretion measurement, and islets were harvested for protein normalization.

### 2.5. Metabolic Hormone Measurement 

Plasma insulin levels were measured using the Mouse Insulin ELISA from Mercodia (Cat: 10-1247-10, Uppsala, Sweden) according to the manufacturer’s instructions and analyzed using the cubic spline. Serum levels of glucose-dependent insulinotropic polypeptide (GIP), insulin, C-peptide, leptin, pancreatic polypeptide (PP), and pro-inflammatory cytokines such as IL-6 and TNF-α were measured using the Milliplex MAP Mouse Metabolic Hormone Magnetic Bead Panel (Millipore, Billerica, MA, USA) following the manufacturers’ instructions. The HOMA-IR index was calculated using the formula: (fasting serum insulin [mU/L] × fasting glucose [mmol/L])/22.5.

### 2.6. RNA Isolation, RT-PCR, and Real-Time PCR

Total RNA from islets was isolated using an Arcturus PicoPure RNA Isolation Kit, Catalog # KIT0202, KIT0204 (ABI), following the manufacturer’s instructions. The cDNA was synthesized from 250–500 ng RNA using the SuperScript III First-Strand Synthesis System (Invitrogen, Carlsbad, CA, USA). Real-time RT-PCR was performed on a Bio-Rad Real-Time PCR Cycler (Bio-Rad Lab., Hercules, CA, USA) using SYBR Green PCR Master Mix according to the protocol provided by the manufacturer. Relative gene expression levels were normalized with 18S rRNA or β-actin. 

The primers used in the qPCR were as follows: Slc2a1: forward primer 5′-TGTGGTGTCGCTGTTTGTTGT-3′, reverse primer 5′-CCTCGGGTGTCTTGTCACTT-3′; Slc2a2: forward primer 5′-ATCATTGGCACATCCTACT-3′, reverse primer 5′-TCAGTTC CTCTTAGTCTCTTC-3′; Insr: forward primer 5′-CAGAAGCACAATCAG AGTGAGTATGAC-3′, reverse primer 5′- ACCACGTTGTGCAGGTAATCC-3′; Irs1: forward primer 5′-GCCTGGAGTATTATGAGAACGAGAA-3′, reverse primer 5′- GGGGATCGA GCGTTTGG-3′; Irs2: forward primer 5′-ACTTCCCAGGGTCCCACTGCTG-3′, reverse primer 5′-GGCTTTGGAGGTGCCACGATAG-3′; Ins1: forward primer 5′-GACCAG CTATAATCAGAGACC-3′, reverse primer 5′-AGTTGCAGTAGTTCTCCAGC TG-3′; Ins-2: forward primer 5′-AGCCCTAAGTGATCCGCTACAA-3′, reverse primer 5′-AGTTGCAGTAGTTCTCCAGCTG-3′; Akt1: forward primer 5′-GACCCACGACC GCCTCTG-3′, reverse primer 5′-GACACAATCTCCGCACCATAGAAG-3′; Prkaa1: forward primer 5′-AAGCCGACCCAATGATATCA-3′, reverse primer 5′-CTTCCTTCGTAC ACGCAAAT-3′; Il1β: forward primer 5′-TGTTCTTTGAAGTTGACGGACCC-3′, reverse primer 5′-TCATCTCGGAGCCTGTAGTGC-3′; Tnf: forward primer 5′-GAGAAAGTCA ACCTCCTCTCTG-3′, reverse primer 5′-GAAGACTCCTCCCAGGTATATG-3′.

### 2.7. Statistical Analysis

Data were presented as mean ± SEM and statistical significance was tested using GraphPad Prism 8.0.1 (GraphPad Software, La Jolla, CA, USA). The normality of the data was tested using the Shapiro–Wilk normality test. For the data that were normally distributed, a one-way analysis of variance was used. For the data that were not normally distributed, the non-parametric Kruskal–Wallis test was performed. Throughout this paper, genotype differences between CTRL and *Ghsr*-βKO mice are denoted with *, and aging effects are denoted with #.

The sample size was determined based on our prior experience with the same mouse colony [29,30]. The exact sample size for each experiment is detailed in the figure legends. In the HFD-induced obesity experiment, each group consisted of 10 animals. For the analysis of metabolic hormones and pro-inflammatory cytokines, we utilized 4–5 mice for the fed condition, and 9 or 5 for the fasting condition. In the in vivo and ex vivo GSIS analysis of aging mice, 5–6 mice were utilized per group. Gene expression profiles of islets were obtained from 4 mice per age group. For the in vivo analysis of the aging effect on β-cell-specific GHSR-deleted mice, 5 aged mice per group were used. For the ex vivo GSIS analysis on aged CTRL and *Ghsr*-βKO mice, we employed 5–6 aged mice per group. In the STZ-induction experiments, 3–4 mice per genotype per age group were utilized for age comparison. 

## 3. Results

### 3.1. β-Cell Ghsr Deficiency Had no Discernable Effect on HFD-Induced Obesity and Glucose Dysregulation

First, we investigated whether β-cell GHSR regulated insulin secretion under HFD-induced obesity. *Ghsr*-βKO and CTRL mice were subjected to HFD for 16 weeks from 8 weeks of age. Bodyweight changes and body composition were monitored throughout the HFD feeding. HFD-fed *Ghsr*-βKO mice did not show any differences in body weight nor fat mass (Figure 1A). *Ghsr*-βKO mice initially exhibited lower fed blood glucose compared with the control mice during the early stages of HFD feeding (Figure 1B). However, after 8 weeks of HFD, no differences in blood glucose between genotypes were observed, indicating that β-cell GHSR did not exert a lasting effect on fed glucose (Figure 1B).

To investigate whether β-cell GHSR regulated insulin secretion in response to glucose load, we conducted GTT and in vivo GSIS on HFD-fed *Ghsr*-βKO and CTRL mice (Figure 1C,D). The blood glucose levels following glucose stimulation were similar between the genotypes (Figure 1C,D). Similarly, plasma insulin levels exhibited no significant difference between the genotypes (Figure 1C,D). The blood glucose levels during the ITT at 12 weeks after HFD feeding also showed no significant difference between genotypes (Figure 1E). Collectively, although the impact of GHSR deficiency was observed during the early stages of HFD administration, it did not significantly alter insulin secretion, glucose intolerance, nor insulin sensitivity in HFD-induced obesity.

### 3.2. Aging Is Correlated with Impaired Insulin Sensitivity and Elevated Inflammatory Cytokines

In addition to obesity, glycemic dysregulation is common in aging as well. To understand the glycemic regulation, insulin sensitivity, and inflammation profile in aging, we examined blood glucose, serum metabolic hormones, and circulating pro-inflammatory cytokines of young (6-to-7-months-old) and aged (20-to-21-months-old) mice and calculated the index of homeostasis model assessment of insulin resistance (HOMA-IR). The fed and fasting blood glucose were significantly decreased in aged mice compared with young mice (Figure 2A). In aged mice, while fed serum insulin showed a decreasing trend, fasting serum insulin showed an increasing trend (Figure 2B), leading to a significantly increased HOMA-IR in aged mice (Figure 2C). The differential changes of fed and fasting insulin levels in aged mice suggest that impaired insulin secretion in aging is dependent on metabolic state. Interestingly, serum C-peptide levels were not altered by aging (Figure 2D), suggesting that changes in insulin levels were not due to insulin production but due to secretion. IL-6, considered a hallmark cytokine of inflamm-aging [42,43], was significantly increased in aged mice in a fed condition (Figure 2E), while TNF-α was not affected (Figure 2E). Collectively, these data showed that aged mice had increased fasting insulin and elevated pro-inflammatory cytokine IL-6, both contributing to the insulin-resistant state in aged mice. 

### 3.3. Aging Impairs Glucose-Stimulated Insulin Secretion (GSIS) Both In Vivo and Ex Vivo

Both insulin secretion and systemic insulin sensitive play a crucial role in glycemic regulation. To assess the effect of aging on β-cell functional responsiveness, we conducted both in vivo and ex vivo GSIS in young and aged mice. The in vivo GSIS was conducted by intraperitoneal injection of the mice with a high dose of 3 g of glucose per kg of body weight. The results demonstrated that aged mice had significantly impaired glucose tolerance compared with young mice (Figure 3A). Despite the impairment of glucose tolerance, the aged mice showed significantly higher plasma insulin levels in both basal and glucose-stimulated conditions (Figure 3B). When normalized to the basal level, the first peak of insulin at 2 min after glucose injection was significantly lower in the aged mice (Figure 3C). This finding indicates that the insulin responsiveness to glucose was less sensitive in aged mice.

To determine the effect of aging on insulin release under a controlled environment, we conducted ex vivo GSIS by treating pancreatic islets with 3.3 mM (basal) and 11.2 mM (hyperglycemic) glucose. Pancreatic islets were isolated from young mice and aged mice. Similar to the in vivo GSIS (Figure 3C), the ex vivo GSIS in islets from aged mice showed reduced insulin secretion in response (Figure 3D). Both in vivo and ex vivo GSIS data demonstrated that aged mice exhibited impaired insulin secretion under hyperglycemic conditions.

### 3.4. Pancreatic Islets Show Differential Changes in Gene Expression during Aging

To determine the underpinning mechanism of impaired GSIS in aged mice, we studied age-associated changes of genes involved in glucose uptake and the insulin signaling pathway in islets that had been isolated from 3-, 10-, and 15-month-old mice. 

The gene Slc2a2 encodes glucose transporter-2 (GLUT-2), and Glut 2 is the primary glucose transporter in insulin-secreting β cells of rodents [44]. We found the expression of Slc2a2 was significantly decreased in the islets of 15-month-old mice compared with those of 3- and 10-month-old mice (Figure 4A). On the other hand, the expression of GLUT-1 gene Slc2a1 was significantly increased in the islets of 10- and 15-month-old mice compared with those of 3-month-old mice (Figure 4A).

Research indicates that insulin signaling significantly influences the functionality of pancreatic islet β cells; when this signaling pathway is compromised, it can lead to β-cell apoptosis and trigger the onset of T2D [31]. So, next, we examined genes related to the insulin signaling pathway, including Insr, Irs1, and Akt1, which encode insulin receptor, IRS-1, and AKT1 kinase, respectively (Figure 4B). We found the expression of genes associated with the insulin signaling pathway gradually decreased during aging, supporting a decrease in insulin sensitivity with age.

Studies have shown that β-cell function is affected by cytokines expressed in the islet microenvironment under stress conditions, including pro-inflammatory cytokines such as TNFα and IL-1β [45]. While the gene expression of TNF and IL-1β was significantly increased in the aged mice, Il1β showed earlier induction compared with Tnf (Figure 4C). The chronological changes in gene expression in the islets are depicted in a schematic diagram in Figure 4D. The Ghrelin-GHSR signaling pathway has been proposed to play a pivotal role in the aging process [46]. Also, we previously reported that the expression of GHSR in adipose tissues increased with aging [35]. In pancreatic islets, unlike adipose tissue, we observed a chronological decrease in the gene expression of Ghsr, which encodes the ghrelin receptor GHSR (Figure 4E).

In summary, gene expression within islets indicated that aging was associated with chronological changes in the glucose transporter, insulin signaling, and inflammatory cytokines. Specifically, aging was correlated with a differential expression of glucose transporter, decreased insulin signaling, and increased expression of inflammatory genes. 

### 3.5. β-Cell Ghsr Deficiency has Little Effect on In Vivo GSIS but Improves Ex Vivo GSIS in Normal Aged Mice

To determine the effect of GHSR in β-cells during normal aging, we next investigated 15-month-old *Ghsr*-βKO and CTRL mice that had been maintained on a regular diet throughout their lifespan. Similar to HFD-fed groups, in vivo GSIS results from aged CTRL and *Ghsr*-βKO groups did not show a significant difference in glucose tolerance (Figure 5A). Likewise, insulin secretion under glucose stimulation did not differ between the two groups in terms of either absolute or normalized values (Figure 5B). Interestingly, ex vivo GSIS revealed a significant difference between the CTRL and *Ghsr*-βKO groups, with *Ghsr*-βKO mice exhibiting higher insulin secretion under both basal and hyperglycemic conditions (Figure 5C). The results indicate that in aging, *Ghsr* deficiency in β cells improves insulin secretion ex vivo but not in vivo.

### 3.6. β-Cell Ghsr Deficiency Protects against STZ-Induced β-Cell Damage in Aged Mice

Ghrelin receptor GHSR, a nutrient sensor, is activated in response to environmental stressors [47]. Since β-cell GHSR had minimal effects under normal aging in vivo, we hypothesized that β-cell GHSR might have a more pronounced effect under STZ-induced β-cell injury. After a single high dose of STZ (150 mg/kg body weight) IP injection, we monitored body weight, blood glucose, and plasma insulin levels, and conducted glucose tolerance testing (GTT) in young, middle-aged, and old CTRL and *Ghsr*-βKO mice. A schematic diagram of the experimental setup is shown in Figure 6A.

As expected, the administration of STZ increased fed blood glucose levels in all age groups (Figure 6B–D). The STZ-induced increase in fed blood glucose levels was significantly higher from day 1 in both young and old groups and from day 3 onward in the middle-aged group, persisting until termination (Figure 6B–D). Intriguingly, the impact of STZ varied with age. In the control mice, the STZ-induced increase in fed blood glucose after 8 days was 3.86-fold greater than for controls in the young group, while it was 2.94-fold greater in the middle-aged and 2.77-fold greater in the old group (Figure 6B–D). Compared with the young group, both the middle-aged and old groups showed significantly lower fed blood glucose with the STZ treatment, suggesting that aged mice have a dampened response to STZ. 

Next, we compared the response to STZ in *Ghsr*-βKO and CTRL mice of different ages. In young mice, the β-cell-specific GHSR deletion did not affect the STZ-induced increase in fed blood glucose or the decrease in plasma insulin (Figure 6B). Similarly, GHSR deletion in β cells did not alter glucose homeostasis in IPGTT (Figure 6B). In middle-aged and old mice, *Ghsr*-βKO significantly suppressed the STZ-induced increase in fed blood glucose (Figure 6C,D). Similarly, the STZ-induced decrease in plasma insulin was attenuated in *Ghsr*-βKO mice in the middle-aged and old groups (Figure 6C,D). Furthermore, *Ghsr*-βKO mice showed improved glucose tolerance with significantly higher insulin secretion at 30 min after glucose injection compared with the CTRL groups (Figure 6C,D). These findings demonstrate that middle-aged and old *Ghsr*-βKO mice were protected from STZ-induced β-cell damage, showing attenuated hyperglycemia and improved glucose tolerance. 

### 3.7. Islets from β-Cell Ghsr-Deficient Mice Exhibit Differential Gene Expressions of Insulin Signaling and Pro-Inflammatory Cytokines

To further assess the genes governing insulin secretion in the pancreatic islets, we conducted gene expression analysis on islets isolated from aged CTRL and *Ghsr*-βKO mice. Under normal aging conditions, while there was no difference in the expression of glucose transporter genes, the islets of aged *Ghsr*-βKO mice showed significantly increased expression of genes related to insulin signaling, including Insr and Irs1 (Figure 7A,B). Also, the islets of aged *Ghsr*-βKO mice displayed reduced expression of Tnf but not Il1β (Figure 7C).

Next, we assessed gene expression in the pancreatic islets of old STZ-treated CTRL and *Ghsr*-βKO mice. Similar to the islets under normal aging conditions, there were no differences in the expression of glucose transporter genes between CTRL and *Ghsr*-βKO mice (Figure 7D). The islets of *Ghsr*-βKO mice exhibited elevated expression of genes associated with insulin signaling, such as Irs1 and Ins1 (Figure 7E). Additionally, old *Ghsr*-βKO mice displayed reduced expression of genes linked to the inflammatory pathway, such as Il1β and Tnf, compared with old CTRL mice (Figure 7F). 

The gene expression difference between old CTRL and *Ghsr*-βKO mice was more pronounced under STZ than normal aging. Aged *Ghsr*-βKO mice had improved insulin signaling and reduced inflammatory gene expression in the islets, well in line with increased insulin secretion in vivo.

## 4. Discussion

Ghrelin, an acylated 28-amino acid peptide, is an appetite-stimulating peptide hormone secreted mainly by the stomach [48]. In the initial stages of ghrelin research, it was discovered that plasma ghrelin levels were inversely correlated with fasting insulin levels [48]. This led to extensive research on the role of ghrelin in insulin secretion and glycemia [48,49]. Our early findings also indicated the involvement of ghrelin signaling in insulin secretion, demonstrated by using ghrelin knockout leptin-deficient mice (*ob*/*ob*), and we found that deletion of ghrelin augmented GSIS and insulin sensitivity [50]. Subsequently, there have been similar observations by others showing ghrelin suppressing insulin secretion in various species, including mice, rats, and humans. [51,52]. While the literature has consistently reported the effects of ghrelin on insulin secretion, there have been confounding observations on the role of GHSR in GSIS. These discrepancies may be attributed to variations in the mouse models used and/or differences in the experimental protocols of GSIS. In a study utilizing the rat insulin promoter-driven Cre recombinase (*Ins-Cre*) model, Kurashina et al. reported that ghrelin attenuated GSIS via β-cell GHSR [53]. However, as that study primarily focused on exploring whether ghrelin attenuated GSIS via β-cell GHSR, it did not comprehensively elucidate whether GHSR also affected GSIS. It is important to investigate the direct effects of GHSR on insulin secretion, as GHSR exhibits substantial constitutive activity through ligand-independent intracellular signaling pathways. We previously reported that pancreatic β-cell GHSR decreased GSIS and improved insulin sensitivity [29]. 

In the current study, we investigated the effects of β-cell GHSR deficiency on glycemic regulation in obesity and aging using *Ghsr*-βKO mice. Under DIO, β-cell GHSR deficiency did not exhibit a notable effect on body weight/fat, fed blood glucose, glucose intolerance, nor insulin resistance (Figure 1). Insulin secretion under hyperglycemic conditions was also unchanged, as shown by in vivo GSIS (Figure 1D). While there have been no studies specifically focused on GHSR, some previous studies have explored the influence of ghrelin signaling on insulin secretion under HFD conditions. For example, blocking the ligand effect of ghrelin through the antagonist and ghrelin knockout prevented HFD-induced glucose intolerance [54]; in the study, antagonism of ghrelin enhanced insulin release without altering insulin sensitivity [54]. Similarly, knockout of ghrelin O-acyltransferase (GOAT), the only enzyme known to catalyze acyl modification of ghrelin, led to improved glucose tolerance under HFD, accompanied by increased insulin secretion under hyperglycemic conditions [55]. However, insulin sensitivity following HFD administration was not tested, which makes it difficult to determine the effect of GOAT on insulin sensitivity [55]. More recently, reducing ghrelin in mice through both germline deletion or conditional cell ablation increased pancreatic islet size by decreasing β-cell apoptosis and increasing β-cell proliferation [56]. Collectively, these studies suggest the effect of ghrelin signaling on insulin secretion. However, in our study, β-cell-specific GHSR deficiency did not alter HFD-induced glucose intolerance or insulin secretion, suggesting differential effects between ghrelin and GHSR under obesity.

The research on the effects of ghrelin signaling on insulin secretion under aging has been more limited than that on DIO. We have reported the significant effects of GHSR in white and brown adipose tissue in aging [35,38,39,40]. It is well established that aging is associated with insulin resistance due to increased adiposity and sarcopenia, making individuals more susceptible to type 2 diabetes and other chronic diseases [57]. However, the insulin secretion aspect of glycemic control has not been as extensively studied in aging, there are conflicting reports on the effects of aging on insulin secretion. Some studies in rodents and humans have reported a decline in islet function with aging [16,58,59], while other studies have shown preserved or even increased GSIS with age [60,61,62,63]. Interestingly, most insulin secretion islet studies in aging have been conducted on human islets ex vivo [16,58,59,60,61,62], and in vivo data are notably lacking. In this study, we conducted a comprehensive set of in vivo and ex vivo studies to assess the function of pancreatic islets in aging.

Our data support the association of aging with glucose intolerance, insulin resistance, and impaired insulin secretion under hyperglycemic conditions, as evident in HOMA-IR and in vivo/ex vivo GSIS (Figure 2 and Figure 3). These findings align with previous studies that have also shown that aging compromises insulin secretion in response to high glucose [16,58,59]. Furthermore, our ex vivo GSIS results demonstrated that islets from aged mice exhibited reduced insulin secretion even at the basal glucose concentration compared with young mice (Figure 4B), which is in line with age-associated insulin resistance. It is important to note that fasting insulin levels in the aged mice may not fully represent the actual insulin secretory function, since aging is known to impair insulin clearance [64]. Overall, our findings support the notion that aged mice display impaired glucose metabolism characterized by both insulin resistance and impaired insulin secretion. These observations highlight the importance of studying both insulin resistance and insulin secretory function in the context of aging.

To explore the underlying mechanism of age-associated impairment of insulin secretion, we studied circulating cytokine and gene expression in pancreatic islets. We detected elevated circulating pro-inflammatory cytokines and alterations in gene expression related to glucose transporters and insulin signaling in the islets (Figure 2 and Figure 4). It has been documented that elevated levels of proinflammatory cytokines in circulation are associated with pancreatic β-cell dysfunction and the development of type 2 diabetes [65,66]. Clinical studies on T2D patients have shown that a decrease in circulating TNF-α levels is associated with improved β-cell function and insulin sensitivity after receiving transient intensive insulin therapy [67]. Similarly, experimental conditions of low-grade systemic inflammation through infusion of circulating IL-1β and IL-6 in prediabetic mice resulted in islet dysfunction and the development of T2D [68]. While the effect of inflammation on β-cell dysfunction and T2D development is not well documented, it is not clear how the aging-induced inflammatory state in older individuals affects insulin secretion. One of the prominent characteristics of aging is the presence of chronic, low-grade inflammation, often referred to as “inflamm-aging.” [69]. This state is characterized by elevated circulating levels of pro-inflammatory cytokines, including IL-6 and TNF-α [69]. Indeed, in our study, we found an age-related increase in serum IL-6, but not TNF-α (Figure 2E). While this result suggests that systemic inflammation may contribute to β-cell dysfunction in aging, further investigations are required to determine whether there is a direct cause-and-effect relationship between elevated islet inflammation and T2D development in aging. 

It has been suggested that inflammation induced by islet-derived inflammatory signaling may play a more prominent role in impairing β-cell function [45]. Therefore, we tested gene expression associated with inflammatory signaling pathways, including IL-1β and TNF-α, and observed a significant increase in the expression of both genes with increased age (Figure 4C). Consistent with our findings, it was observed that aging was linked to elevated gene expression of IL-1β, but not IL-6, in islets [58]. Moreover, both the global and myeloid cell-specific deletions of IL-1β protect islets from the age-associated decline in β-cell function [58], highlighting the significance of the IL-1β signaling pathway in the aging islets. Our findings suggest that impaired insulin secretion in aging is probably linked to an increase in circulating inflammatory cytokines and an age-related increase of pro-inflammatory genes in the islets. These observations support the notion that inflammatory state, both systemically and within the islets, may play a role in impairment of β-cell function and the development of T2D in aging.

In addition, we found that genes related to glucose transporters and the insulin signaling pathway were altered in the islets of aged mice (Figure 4A,B). In rodent pancreatic β-cells, GLUT-2 is essential for glucose-stimulated insulin secretion, and studies with GLUT-2 knockout (*Slc2a2^−^*^/*−*^) mice have shown impaired glucose clearance and insulin secretion [44]. Islets isolated from GLUT-2 knockout mice exhibited increased glucose uptake at low glucose but no further increase at high glucose, indicating impaired insulin response under hyperglycemic conditions [44]. Also, many rodent diabetes models exhibiting similar symptoms to human diabetes mellitus had decreased GLUT-2 expression [44]. In this study, we observed a decrease in GLUT-2 mRNA expression in 15-month-old mice compared with both 3-month-old and 10-month-old mice (Figure 4A). Our results are consistent with previous findings showing reduced gene expression of GLUT-2 in older animals [60,70,71,72]. GLUT-1 is not the most abundant glucose transporter in pancreatic β cells, but studies have shown its involvement in insulin secretion [73,74]. For example, re-expression of GLUT-1 in the pancreatic β-cells of GLUT-2-null mice rescued the mice from lethality and restored normal insulin levels in hyperglycemic clamp assessment [74]. Interestingly, β-cell-specific deletion of insulin-degrading enzyme increased GLUT-1 by 60%, leading to enhanced glucose transport and insulin secretion under low-glucose condition [75]. However, the age-related alteration in GLUT-1 expression in the islets has not been thoroughly explored. Our results indicated an increase in GLUT-1 expression in islets with aging (Figure 4A), supporting the possibility of compensatory GLUT-1 expression in response to reduced GLUT-2 expression in aging.

Insulin is not only a metabolic regulator, but also plays a pivotal role in proliferative signaling pathways in the pancreatic islets through autocrine feedback [76]. Studies have shown that insulin signaling has a significant role in controlling the function of pancreatic islet β cells; when insulin is compromised, it can result in β-cell apoptosis and trigger the onset of T2D in obese subjects [76]. However, the role of insulin signaling in aging islets has yet to be investigated. We studied genes associated with the insulin signaling pathway, and we found that IR, IRS1, Akt-1, and Ampka1 were reduced in the islets of aged mice (Figure 4B). We believe this is the first report indicating that aging negatively regulates insulin signaling in the islets.

Here, we also investigated the effects of β-cell GHSR deficiency under normal aging, accompanied by chronic low-grade inflammation, insulin resistance, and impaired insulin secretion. Similar to HFD-treated *Ghsr*-βKO mice, aged *Ghsr*-βKO mice did not exhibit a significant decrease in in vivo GSIS (Figure 5A–C). Intriguingly, ex vivo GSIS under both basal and hyperglycemic conditions was increased in aged *Ghsr*-βKO mice compared with CTRL (Figure 2D). Though limited, our data suggest that β-cell GHSR may influence insulin secretion during aging, and β-cell GHSR regulates insulin secretion under acute injury in aging.

STZ is widely used in experimental animals to induce type 1 diabetes mellitus, and it is also used to induce inflammation in pancreatic islets to mimic pancreatic insulitis [77,78]. It is known that pancreatic β cells selectively uptake STZ through low-affinity GLUT2, and cells lacking GLUT2 are immune-protective to STZ insult [79,80]. Aging increases vulnerability to diabetes, so we studied β-cell-GHSR-deficient mice under STZ-induced hyperglycemia. We challenged young, middle-aged, and old *Ghsr*-βKO mice using a single high dose of STZ to induce acute β-cell injury. A single injection of high-dose STZ (60–65 mg per kg body weight) has been shown to induce inflammation through oxidative stress [81,82,83]. We found that aged mice were less responsive to STZ, so we utilized a single injection of a higher dose of STZ (150 mg per kg body weight) to induce pancreatic inflammation. In our study, young mice had more severe STZ-induced hyperglycemia compared with the middle-aged and old mice (Figure 6). Others have reported that the age of the animals affects the efficiency of STZ, as STZ induced more severe β-cell destruction in 2-month-old young mice than in 10-month-old mice [84]. This could be attributed to the age-induced decrease in GLUT2 expression in the aged mice, as demonstrated by us (Figure 5A) and others [60,70]. 

Intriguingly, under high-dose STZ challenge, *Ghsr*-βKO mice exhibited significantly attenuated hyperglycemia, showing improved glucose tolerance and increased insulin secretion in middle-aged and old mice, but not in young mice (Figure 5). Furthermore, *Ghsr*-βKO mice were protected from STZ-induced impairment of insulin signaling and the upregulation of pro-inflammatory cytokines compared with CTRL mice (Figure 7). This improvement was probably not due to a difference in GLUT2 expression, as *Ghsr*-βKO had comparable GLUT2 expression compared with their control counterparts (Figure 7). The enhanced insulin secretory capacity in *Ghsr*-βKO is likely to have resulted from suppressed inflammation and/or improved insulin sensitivity in the pancreatic islets (Figure 7). 

Chronic exposure to pro-inflammatory mediators has been shown to activate inflammatory signaling pathways, leading to the inhibition of the insulin signaling pathway in β cells of pancreatic islets [85,86]. IL-1-induced impairment of the insulin signaling pathway leads to β-cell dysfunction, apoptosis, and ultimately development of T2D [85]. Our data suggest that reduced inflammatory response and upregulated insulin signaling-related gene expression may contribute to the preservation of β-cell function in aged *Ghsr*-βKO mice. Interestingly, the increased gene expression related to the insulin signaling pathway in the islets of aged *Ghsr*-βKO mice was preserved under both normal physiological conditions and STZ-induced stress conditions in aging (Figure 7). Also, aged *Ghsr*-βKO mice exhibited improved in vivo insulin secretion only under STZ (Figure 6). Collectively, our results indicate that GHSR plays a crucial role in β-cell function in aging, particularly under stress conditions.

Overall, our findings suggest that β-cell GHSR plays a crucial role in β-cell function and the development of diabetes in aging. Suppressing GHSR in β-cells may offer a novel therapeutic strategy for preserving β-cell resilience and improving β-cell function, protecting against inflammation and insulin resistance in the aging pancreatic islets. To the best of our knowledge, this is the first set of studies to highlight the significance of GHSR in β cells in aging. While the prevalence of diabetes and impaired glucose tolerance is generally higher in males compared with females, it is important to note that the prevalence of diabetes is higher in females of obese older populations. There is sexual dimorphism in the development of diabetes. In this study, we studied only male mice; therefore, we cannot assume that β-cell GHSR deficiency would have the same effect in females. Excitingly, it has been shown that GHSR expression in islets is positively correlated with age in humans, suggesting that our findings about β-cell GHSR in aging may also apply to human biology [87]. However, whether our results are relevant to humans needs further investigation. 

## 5. Conclusions

Our study investigated pancreatic β-cell function under chronic inflammation in DIO and aging, as well as the resilience of β-cells to acute inflammatory injury during aging. We studied β-cell GHSR deficiency under DIO and aging and found that β-cell GHSR deficiency did not significantly alter insulin secretion or glucose intolerance. Similarly, aged *Ghsr*-βKO mice exhibited comparable in vivo GSIS to the CTRL group under normal physiological conditions. Interestingly, aged *Ghsr*-βKO mice showed improved insulin secretion under hyperglycemic conditions compared with their control.

Also, we characterized the phenotypes of aged mice in terms of insulin sensitivity and secretion. Aged mice had higher levels of circulating inflammatory cytokines and impaired GSIS both in vivo and ex vivo. Furthermore, aging resulted in altered gene expression in islets, characterized by shifts in glucose transporter expression, diminished insulin signaling, and increased expression of inflammatory genes. 

To examine the effects of β-cell GHSR under STZ-induced β-cell injury, we challenged young, middle-aged, and old *Ghsr*-βKO mice with STZ. Remarkably, aged *Ghsr*-βKO mice showed a significant protective effect, exhibiting reduced hyperglycemia under STZ treatment. In addition, β-cell GHSR deficiency enhanced insulin signaling and decreased inflammatory cytokines in the islets in aged mice. Overall, our findings suggest that β-cell GHSR could be a potential therapeutic target for improving insulin secretion by modulating inflammation in β cells during aging (Figure 8).

## Figures and Tables

**Figure 1 nutrients-16-01464-f001:**
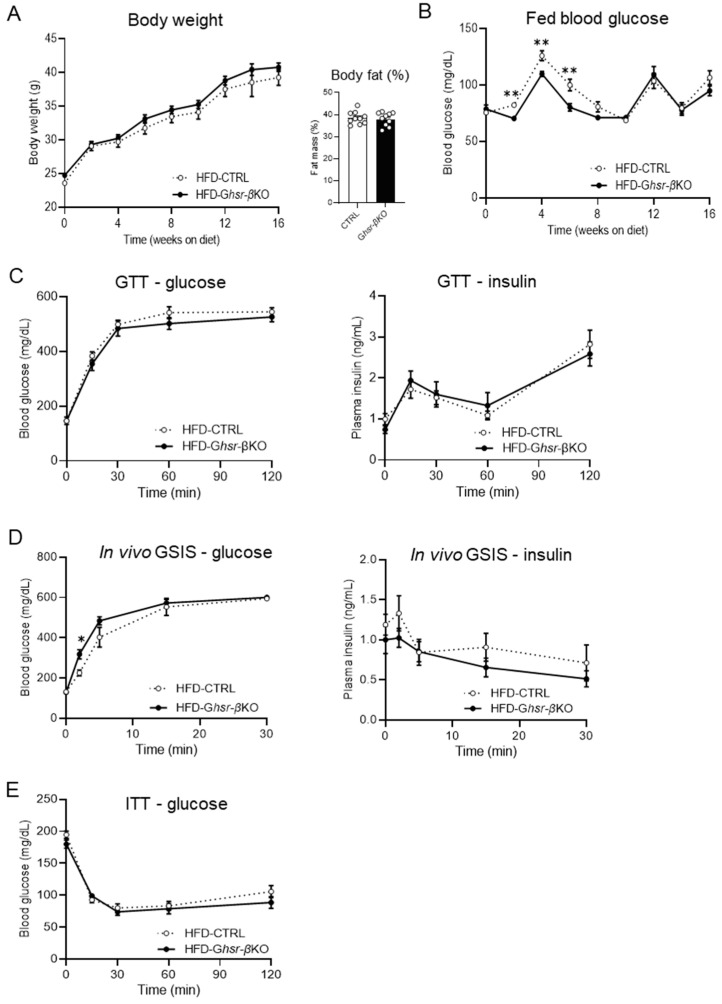
Body composition and glucose phenotype of HFD-fed β-cell-specific GHSR-deleted mice. In this study, 8-week-old CTRL and *Ghsr*-βKO mice were fed with HFD for 16 weeks. Body weight, fat mass, and fed blood glucose were conducted, and various tests including GTT, GSIS, and ITT were applied throughout the study. (**A**) Body weight, fat mass, and (**B**) fed blood glucose after 16 weeks of HFD feeding. (**C**) HFD-fed CTRL and *Ghsr*-βKO mice were subjected to 2 g/kg body weight of glucose for glucose tolerance testing (GTT). GTT was conducted 14 weeks after HFD feeding. Blood glucose levels and plasma insulin are shown. (**D**) In vivo GSIS assay was conducted with IP injection of 3 g glucose per kg body weight. Blood glucose levels and plasma insulin are shown. GSIS was conducted 18 weeks after HFD feeding. (**E**) Insulin tolerance testing was conducted 12 weeks after HFD feeding. n = 10. The data are presented as mean ± SEM, *, and ** representing *p* < 0.05, and *p* < 0.01, respectively, *Ghsr*-βKO compared with the CTRL. (**B**) Fed blood glucose in week 2 (*p* = 0.002), week 4 (*p* = 0.005), week 6 (*p* = 0.005) (ANOVA), (**D**) 2 min after glucose injection (*p* = 0.014) (ANOVA). GTT, glucose tolerance test; GSIS, glucose-stimulated insulin secretion; ITT, insulin tolerance test.

**Figure 2 nutrients-16-01464-f002:**
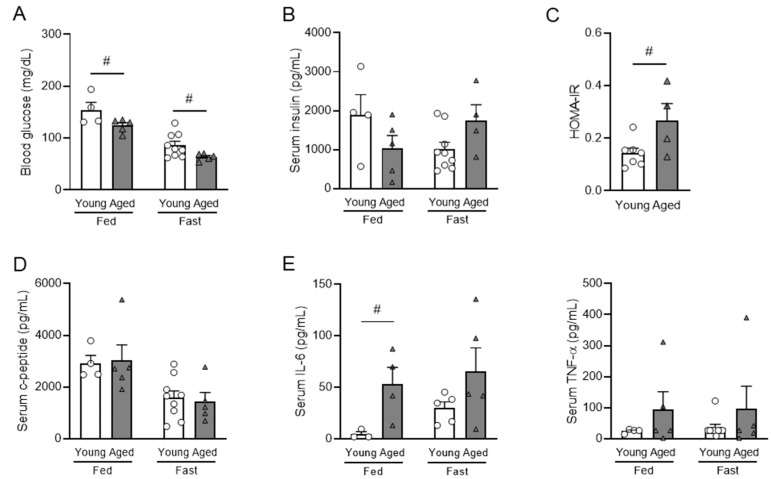
Metabolic hormones and pro-inflammatory cytokines in fed and fasted young and aged mice. Young (6-to-7-month-old) and aged (20-to-21-month-old) male mice were used. The levels of fed and fasting blood glucose (**A**), serum insulin (**B**), and homeostasis model assessment of insulin resistance (HOMA-IR) index (**C**) of young and aged male mice. C-peptide (**D**) and pro-inflammatory cytokines of IL-6 and TNF-α (**E**) of young and aged male mice. n = 4–5 for fed condition, n = 9 or 5 for fasting condition. Data are presented as mean ± SEM. Individual data points are represented by circles (Young) and triangles (Aged). #, *p* < 0.05, aged vs. young. (**A**) blood glucose under fed (*p* = 0.038) (ANOVA) and fasting (*p* = 0.028) conditions (Kruskal–Wallis). (**C**) (*p* = 0.022) (ANOVA). (**E**) (*p* = 0.027) (Kruskal–Wallis). IL-6, interleukin 6; TNF-α, tumor necrosis factor alpha.

**Figure 3 nutrients-16-01464-f003:**
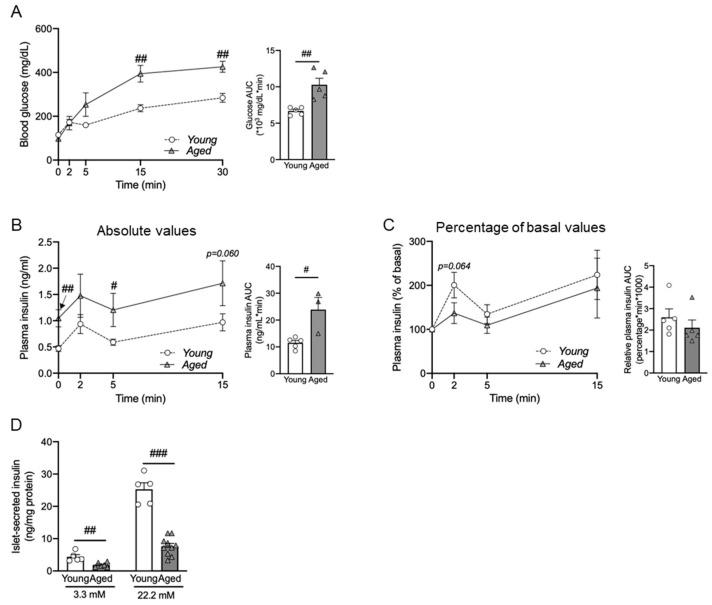
Impaired in vivo and ex vivo glucose-stimulated insulin secretion (GSIS) of aged male mice. For the in vivo GSIS assay, IP injection of 3 g glucose per kg body weight was conducted in young (7-months-old) and aged (15-months-old) male mice. The results show blood glucose and AUC of blood glucose (**A**), absolute values for plasma insulin levels and AUC of plasma insulin (**B**), and relative insulin fold change normalized to baseline at 0 min with corresponding AUC values (**C**). (**D**) Ex vivo GSIS was performed in isolated pancreatic islets from young (5-to-6-months-old) and aged mice (25-to-28-months-old), and insulin secretion was measured. (**A**–**C**) n = 5, (**D**) n = 5–6. The data are expressed as mean ± SEM. Individual data points are represented by circles (Young) and triangles (Aged). #, *p* < 0.05; ##, *p* < 0.01; ###, *p* < 0.001 aged vs. young. (**A**) blood glucose 15 min (*p* = 0.003) and 30 min (*p* = 0.001) after glucose injection (ANOVA), and AUC (*p* = 0.002) (Kruskal–Wallis). (**B**) Plasma insulin baseline (*p* = 0.005), 5 min (*p* = 0.021) after glucose injection (ANOVA), and AUC (*p* = 0.007) (Kruskal–Wallis), (**D**) 3.3 mM (*p* = 0.002), 22.2 mM (*p* < 0.001) (ANOVA).

**Figure 4 nutrients-16-01464-f004:**
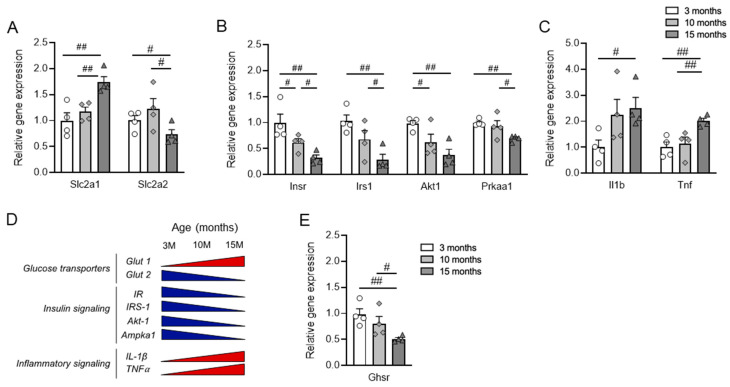
Gene expression profiles of islets from 3-, 10-, and 15-month-old mice. RNA was isolated from the islets of the mice and relative gene expression was analyzed. (**A**) Glucose transporter genes of glucose transporter Slc2a1 and Slc2a2. (**B**) Genes of insulin signaling pathway: Insr, Irs1, Akt1, and Prkaa1. (**C**) Inflammatory cytokines of Il1b and Tnf. (**D**) Schematic diagram of the chronological change of gene expression during aging. The blue triangles denote an age-associated decrease in gene expression, while the red triangles denote an age-associated increase in gene expression. (**E**) Expression of Ghsr in pancreatic islets from 3-, 10- and 15-month-old mice. n = 4. The data are represented as mean ± SEM. Individual data points are represented by circles (3-month-old), rhombuses (10-month-old) and triangles (15-month-old). #, *p* < 0.05; ##, and *p* < 0.01. (**A**) Slc2a1, 3 M vs. 15 M (*p* = 0.004), 10 M vs. 15 M (*p* = 0.004); Slc2 a2, 3 M vs. 15 M (*p* = 0.045), 10 M vs. 15 M (*p* = 0.033). (**B**) Insr, 3 M vs. 10 M (*p* = 0.045), 3 M vs. 15 M (*p* = 0.005), 10 M vs. 15 M (*p* = 0.011); Irs1, 3 M vs. 15 M (*p* = 0.002), 10 M vs. 15 M (*p* = 0.048); Akt1, 3 M vs. 10 M (*p* = 0.034), 3 M vs. 15 M (*p* = 0.002); Prkaa1, 3 M vs. 15 M (*p* = 0.004), 10 M vs. 15 M (*p* = 0.047). (**C**) Il1 b, 3 M vs. 15 M (*p* = 0.012); Tnf, 3 M vs. 15 M (*p* = 0.002), 10 M vs. 15 M (*p* = 0.009). (**E**) Ghsr, 3 M vs. 15 M (*p* = 0.003), 10 M vs. 15 M (*p* = 0.038) (Kruskal–Wallis). If not specified, ANOVA was used. Slc2a1, solute carrier family 2 member 1; Slc2a2, solute carrier family 2 member 2; Insr, insulin receptor; Irs1, insulin receptor substrate 1; Akt1, AKT serine/threonine kinase 1; Prkaa1, protein kinase AMP-activated alpha 1 catalytic subunit; Il1b, interleukin 1 beta; Tnf, tumor necrosis factor; Ghsr, growth hormone secretagogue receptor.

**Figure 5 nutrients-16-01464-f005:**
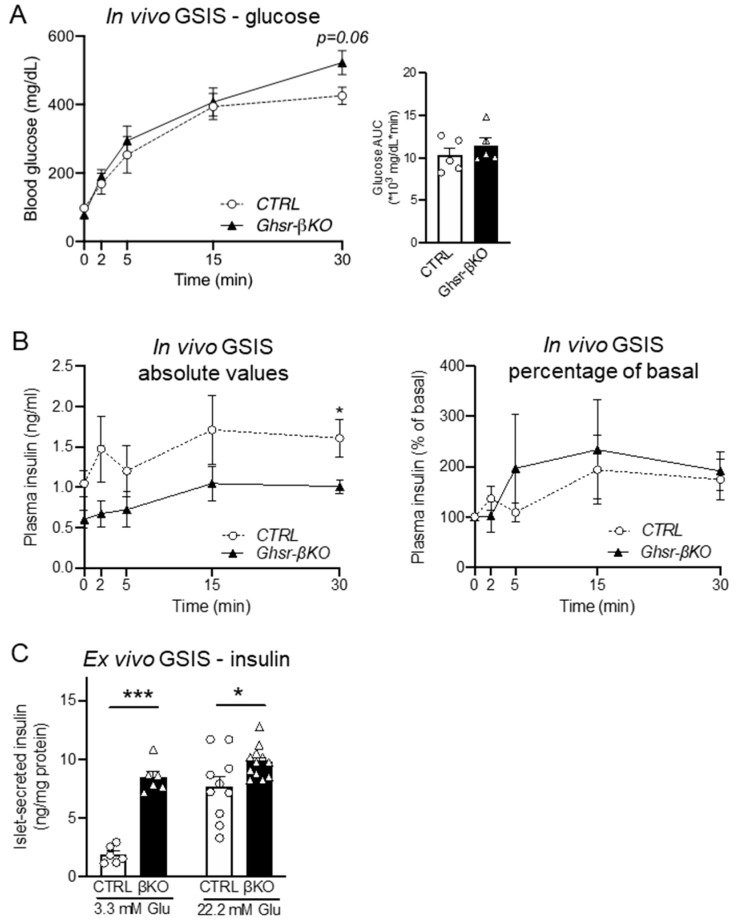
Glycemic phenotype of β-cell-specific GHSR-deleted mice. In vivo GSIS assay with IP injection of 3 g glucose per kg body weight was conducted in aged-matched CTRL and Ghsr-βKO mice at 14.5 months of age. (**A**) Blood glucose and AUC; (**B**) plasma insulin levels and AUC (left), and in vivo insulin fold change normalized to the baseline (right). (**C**) Ex vivo GSIS was conducted in isolated pancreatic islets of aged male CTRL and *Ghsr*-βKO mice (25 to 28 months old), and insulin secretion was measured. The effects of basal (3.3 mM) and high (22.2 mM) glucose concentrations on insulin secretion were measured and normalized to protein content. n = 5–6. The data are presented as mean ± SEM. Individual data points are represented by circles (CTRL) and triangles (*Ghsr*-βKO). * and *** representing *p* < 0.05 and *p* < 0.001, respectively, *Ghsr*-βKO vs. CTRL. (**C**) 3.3 mM (*p* < 0.001) (ANOVA), 22.2 mM (*p* < 0.016) (Kruskal–Wallis). GSIS, glucose-stimulated insulin secretion.

**Figure 6 nutrients-16-01464-f006:**
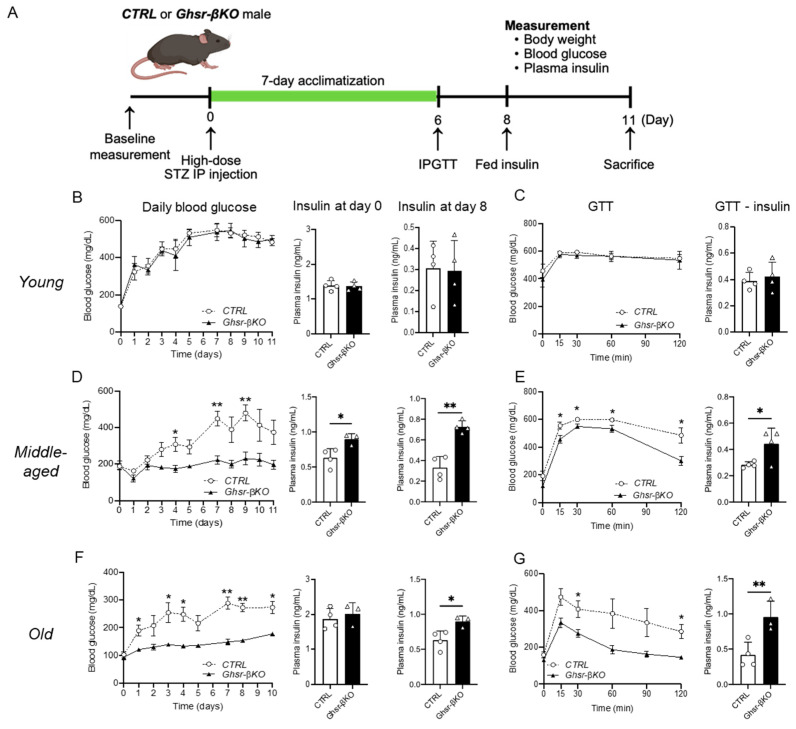
STZ-induced hyperglycemia and impaired glucose tolerance in young, middle-aged, and old β-cell-GHSR-deleted mice. Young (5-to-7-month-old), middle-aged (15-to-17-month-old), or old (25-to-27-month-old) male mice were administrated a single-time high-dose STZ injection. After 7 days of acclimatization, mice were subjected to 1 g/kg body weight of glucose for GTT. (**A**) Schematic diagram of the experimental design is presented. (**B**,**D**,**F**) Daily fed blood glucose, fed plasma insulin on day 0 and day 8 are shown for young (**B**), middle-aged (**D**), and old mice (**F**). (**C**,**E**,**G**) Glucose levels before and after GTT, and insulin secretion at 30 min after glucose injection; results are shown for young (**C**), middle-aged (**E**), and old mice (**G**). n = 3–4 *, *p* < 0.05; **, *p* < 0.01, *Ghsr*-βKO mice vs. CTRL. (**D**) Insulin at day 0, (*p* = 0.025); insulin at day 8, (*p* = 0.008). (**E**) GTT insulin, (*p* = 0.019). (**F**) Insulin at day 8, (*p* = 0.013). (**G**) GTT insulin, (*p* = 0.008) (ANOVA). GTT, glucose tolerance test; STZ, streptozotocin.

**Figure 7 nutrients-16-01464-f007:**
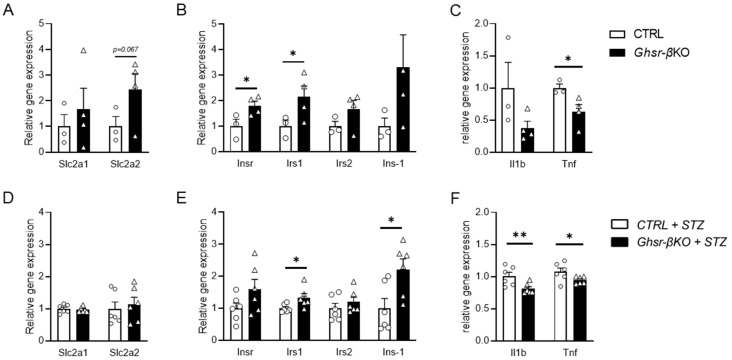
Gene expression in the pancreatic islets from old *Ghsr*-βKO under normal aging and after STZ-induced β-cell injury. RNA was isolated from the mouse islets and relative gene expression was analyzed. mRNA expression of glucose transporter (**A**,**D**), insulin-associated genes (**B**,**E**), and inflammatory cytokines (**C**,**F**) in pancreatic islets of untreated (**A**–**C**) or STZ-treated (**D**–**F**) old CTRL and *Ghsr*-βKO mice (25 to 27 months old). (**A**–**C**) n = 3–4. (**D**–**F**) n = 6. The data are presented as mean ± SEM. Individual data points are represented by circles (CTRL) and triangles (*Ghsr*-βKO). *, and ** representing *p* < 0.05 and *p* < 0.01, respectively, *Ghsr*-βKO vs. CTRL. (**B**) Insr, (*p* = 0.025); Irs1, (*p* = 0.046). (**C**) Tnf, (*p* = 0.024). (**E**) Irs1, (*p* = 0.019); Ins-1 (*p* = 0.011). (**F**) Il1b, (*p* = 0.009); Tnf (*p* = 0.034). ANOVA. Slc2a1, solute carrier family 2 member 1; Slc2a2, solute carrier family 2 member 2; Insr, insulin receptor; Irs1, insulin receptor substrate 1; Akt1, AKT serine/threonine kinase 1; Prkaa1, protein kinase AMP-activated alpha 1 catalytic subunit; Il1b, interleukin 1 beta; Tnf, tumor necrosis factor.

**Figure 8 nutrients-16-01464-f008:**
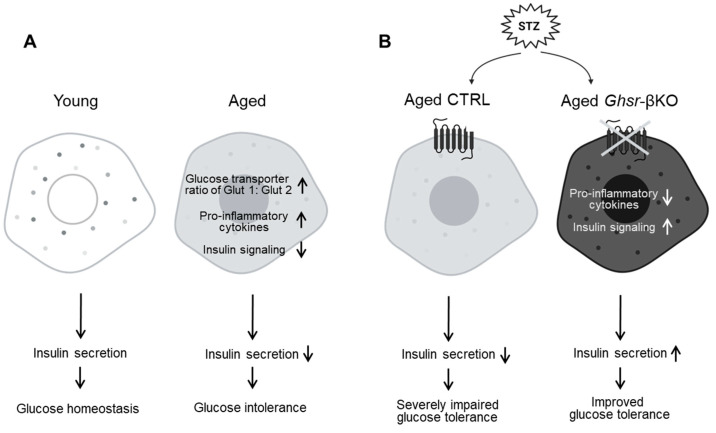
Schematic diagram illustrating the changes in pancreatic islets associated with aging; GHSR deficiency protects against streptozotocin (STZ)-induced β-cell injury in aged mice. (**A**) Normal aging is associated with a switch of glucose transporter 1 to 2, increased pro-inflammatory cytokines, and decreased insulin signaling, resulting in decreased insulin secretion and increased glucose intolerance in aging. (**B**) Under STZ-induced hyperglycemia, deficiency of GHSR in the β-cells is associated with reduced pro-inflammatory cytokines and increased expression of genes related to insulin signaling, resulting in enhanced insulin secretion and improved glucose tolerance.

## Data Availability

The original contributions presented in the study are included in the article, further inquiries can be directed to the corresponding author.

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
