# Peer review of "GHSR Deletion in β-Cells of Male Mice: Ineffective in Obesity, but Effective in Protecting against Streptozotocin-Induced β-Cell Injury in Aging"

_nutrients, 2024, doi:10.3390/nu16101464_

Round 1
Reviewer 1 Report
Comments and Suggestions for Authors
This is a well-written manuscript on the role of GHSR in beta-cells. The study investigates the influence of diet-induced obesity, age and STZ injury, however it only includes males. The sex of the animals are not included in the methods section and the lack of females is not mentioned as a limiting factor and potential differences are not discussed.
Title & Abstract
· This study only includes males, I think that this should be reflected in the title and abstract.
Methods
· The sex of the mice used is not stated in the methods section. I can see further down that only males were used. Why were females not included?
· Why was this method of KO used? As tamoxifen estrogenic effects and estrogens play a role in both body weight regulation and glucose regulation it doesn’t feel like the optimal choice.
· How many subjects were used per group in each experiment?
· How was body composition measured?
· Check your abbreviations, write them out the first time that the word is used, and if abbreviated use throughout text. For example IP.
· Section 2.2 Where and how was blood collected for glucose and insulin measurements? And how was pancreas and serum collected on day 11? Maybe make it clear that this is the sac day for this experiment.
· Section 2.3 How was the blood sampled?
· Statistics largely missing throughout results section: Please write out the statistics, F-value, exact p-values, effect size etc.
Results
· Section 3.1 I think that there’s an extra period in the title.
· Section 3.2 I would remove the first sentence explaining Figure 1B, the second one better describes the initial differences seen. I would also suggest changing the sentence to indicate that there were no long-lasting effects on blood glucose.
· Figure 1 Odd lettering in figure, B missing for example. It is also stated that there are AUC values for GTT and ITT but I do not see these graphs.
· Figure legends in general: I think that it’s better if the figure legend gives a short explanation of what is happening in the graphs rather than just stating what it is and some of the methodology. Please add the number of subjects per group.
· Figure 2, lettering somewhat cut off over graphs.
· Figure 3D is cut off.
· Figure legend 4. It’s always nice to add in the explanations for the abbreviations in the figure legend to make it easy for the reader.
· Figure 5. The “C” is missing over the graph.
· Why are * used in certain graphs to indicate significance and # in others?
· Figure 6. I think that you should further divide this graph in letters to make it easier to understand which graph that you’re referring to in the text. You could also consider adding certain graphs (that you have not mentioned in the text) to the supplementary.
· Section 3.6 “As expected, the administration of STZ increased fed blood glucose levels in all age groups (Figure 6B-D).” Are you looking at the daily blood glucose graphs here? Have you run a statistical test to compare to baseline or are you comparing to a group which didn’t receive the STZ injection?
· Section 3.6. “Intriguingly, the impact of STZ varied with age, as the STZ -induced increase of blood glucose differed with age.” This sentence is a bit double.
· Section 3.6 “Intriguingly, the impact of STZ varied with age, as the STZ -induced 345 increase of blood glucose differed with age. In the control mice, the STZ-induced increase 346 in fed blood glucose after 8 days was 3.86-fold greater compared to controls in the young 347 group, while it was 2.94-fold greater in the middle-aged and 2.77-fold greater in old 348 groups (Figure 6B-D), suggesting that aged mice have dampened response to STZ.” As these are not in the same graph, I am wondering whether you compared them statistically.
· I think Figure 7 is mislabeled. It is Figure 6 and then Figure 5 again.
Discussion
· “This lead to extensive research on the role of ghrelin in insulin secretion and glycemia [48].” You have cited this paper in the previous sentence. I feel like you could either leave this open or cite a review.
Could you add the effect of HFD in the schematic diagram?
Author Response
Dear Reviewer,
We sincerely appreciate your time and effort in reviewing our manuscript entitled “GHSR Deletion in β-Cells of Male Mice: Ineffective in Obesity, but Effective in Protecting Against Streptozotocin-Induced β-Cell Injury in Aging” (ID: nutrients-2989501). Your constructive and insightful comments are greatly appreciated, which has helped to improve our manuscript tremendously. We have carefully addressed all comments and revised the manuscript accordingly. The point-by-point responses to the reviewer’s comments are attached.

Reviewer 2 Report
Comments and Suggestions for Authors
Very interesting manuscript. The role ofinsulin in metabolic diseases is importantespecially in diabetes, obesityand aging. The stimulating effect of GHSR is ofsignificant significance for this type ofstudy. Understanding themechanisms ofdeficiency will lead to better treatment inhumans in the future, which in turn will slowdown thedevelopment of hyperglycemia, obesity and aging processes. It is worthintroducing a model of studiesamongpatients in different age groups, along withthe introduction of changes that couldinhibit this deficiency. Graphics presentedclearly. Literature enough.
Author Response
Dear Reviewer,
We sincerely appreciate your time and effort in reviewing our manuscript entitled “GHSR Deletion in β-Cells of Male Mice: Ineffective in Obesity, but Effective in Protecting Against Streptozotocin-Induced β-Cell Injury in Aging” (ID: nutrients-2989501). Your constructive and insightful comments are greatly appreciated, which has helped to improve our manuscript tremendously.
Response: We sincerely appreciate the reviewer’s positive comments of our study. We concur that exploring the impact of GHSR in chronic inflammatory conditions such as obesity and aging holds significant relevance to diabetic in humans. We have observed an increase in GHSR expression in macrophages during inflammatory states such as obesity and aging. This study showed that GHSR expression within the islets increases with age, underscoring the significance in aging. Intriguingly, our study reveals the pronounced effect of GHSR in b-cell function in aging but not obesity, suggesting the unique role of GHSR in aging diabetes. We firmly believe that our investigation sheds light on the therapeutic implications of ghrelin signaling in aging-associated beta cell dysfunction, GHSR may serve as a promising therapeutic target for diabetes in elderly.
